# Characteristics and Risk Factors for Ischemic Ovary Torsion in Children

**DOI:** 10.3390/children9020206

**Published:** 2022-02-06

**Authors:** Jason Tsai, Jin-Yao Lai, Yi-Hao Lin, Ming-Han Tsai, Pai-Jui Yeh, Chyi-Liang Chen, Yi-Jung Chang

**Affiliations:** 1Department of Neuroscience, Brown University, Providence, RI 02912, USA; jason_tsai@brown.edu; 2Department of Pediatric Surgery, Chang Gung Memorial Hospital, Taoyuan 333, Taiwan; jylai@cgmh.org.tw; 3College of Medicine, Chang Gung University, Taoyuan 333, Taiwan; lin@cgmh.org.tw (Y.-H.L.); drtsai1208@gmail.com (M.-H.T.); 4Department of Obstetrics and Gynecology, Chang Gung Memorial Hospital, Taoyuan 333, Taiwan; 5Department of Pediatrics, Chang Gung Memorial Hospital, Taoyuan 333, Taiwan; charlie01539@hotmail.com; 6Molecular Infectious Disease Research Center, Chang Gung Memorial Hospital, Taoyuan 333, Taiwan; dinoschen@cgmh.org.tw

**Keywords:** ovarian torsion, children, predictive score, ovary, torsion

## Abstract

Identifying ischemic ovary as a complication of ovary torsion (OT) is a significant challenge in children. This study identified risk factors for ischemic OT among pediatric OT patients to prevent delayed treatment. This retrospective study included pediatric inpatients who underwent operation for OT over 20 years. We employed multivariable logistic regression to find the risk factors associated with ischemic OT. Among the 118 patients included in this study, 78 (66.1%) had ischemic OT. Patients with ischemic OT tended to be younger; had more frequent vomiting; and had elevated White blood cell (WBC), C-Reactive protein (CRP), and segments in comparison with non-ischemic OT patients. Multivariable regression showed increased odds of ischemic ovary torsion, associated with higher WBC (12.3 × 10^3^/mm^3^ vs. 8.7 × 10^3^/mm^3^, *p* < 0.001), CRP (50.4 mg/L vs. 8.4 mg/L, *p* < 0.001), and vomiting (55.1% vs. 25%, *p* = 0.002) than in non-ischemic patients. A receiver-operating characteristic (ROC) analysis indicated that patients with vomiting, leukocytosis, or CRP ≧ 40 mg/L were more likely to have ischemic OT (sensitivity, 92%; specificity, 54%; PPV, 79.6; NPV, 78.9%). Ischemic OT is common among pediatric OT patients. The presence of potential risk factors of vomiting, leukocytosis, and CRP more significant than 40 mg/L may assist clinicians in ensuring an expedited surgical treatment.

## 1. Introduction

Ovarian torsion (OT) is a common gynecologic surgical emergency that can afflict women of all ages [1]. Children and adolescent girls account for 15% of all ovarian torsions [2]. The prevalence in females aged 1–20 years is approximately 5 per 100,000 [3]. In ovary torsion, the ovary rotates around its ligamentous support. The ovary’s vascular supply becomes impaired over time, resulting in ovarian infarction or other morbidities, including peritonitis, hemorrhage, sepsis, or adhesion [4,5]. Potential reasons for oophorectomy include suspicion of underlying malignancy or ischemic ovary [6].

The best way to avoid functional repercussions on the ovary is prompt diagnosis and timely detorsion [7]. However, making a diagnosis of OT in the pediatric population remains challenging. According to a meta-analysis, diagnosing ovarian torsion is tricky, with a median delay of 100 h [8]. Recent trends focused on the value of ultrasound and Doppler imaging emergencies [9,10,11]. Ultrasound has 70% and 87% sensitivity and specificity, respectively, which may help with diagnosis [11]. The ovary’s asymmetric growth, the follicles’ peripheral placement, and the ovary’s midline position are all well-documented sonographic hallmarks of intorsion in children, but none are pathognomonic [12]. A lack of Doppler US signals is associated with torsion, but this finding lacks reliability since the literature indicates that 20–73% of torted ovaries lack Doppler flow [13]. In a multicenter study, the rates of oophorectomy for children with ovarian torsion in U.S. children’s hospitals were up to 22.5% [6]. Most investigations indicated that the low salvage rate was related to diagnostic and surgical delays caused by the nonspecific ovarian torsion symptom [14,15].

Developing a diagnostic tool to assist physicians in evaluating ischemic OT would be essential for sustaining ovarian function and fertility in the future. Nevertheless, the low sensitivity, the lack of Doppler flow, and the confusion with hemorrhagic cysts make the overall sensitivity for US diagnosis still restricted [4]. To date, clinical or biological findings that help in predicting the preoperative diagnosis of ischemic OT in children and adolescents remain limited. A comprehensive understanding of the clinical features and the determination of risk variables of ischemic OT are critical since they can assist in focusing on at-risk patients and ensure an expedited surgical treatment. The goal of this research was to find risk factors for ischemic OT in the pediatric OT population and to define clinical characteristics in order to prevent delayed diagnosis, which is imperative in maintaining ovarian function.

## 2. Materials and Methods

### 2.1. Study Design Setting

A retrospective chart review was performed for patients admitted to the Chang Gung Memorial Hospital over 20 years (January 2000 to December 2019), with a diagnosis code of 620.5 for torsion of the ovary, ovarian pedicle, or fallopian tubes, which was later verified by surgery. The Institutional Review Board of the Chang Gung Memorial Hospital approved this study. Direct visibility of a rotated ovary during surgical assessment provided a definite diagnosis of ovarian torsion. The appearance of an ischemic ovary in the surgeon’s operation note was classed as the definition of ischemic OT.

### 2.2. Selection of Participants

We excluded cases of prenatal ovarian torsion because the condition has significant differences in presentation and physical examination. Patients older than 17 were excluded from the analysis since the study aimed to evaluate ovarian torsion in the pediatric population. Clinical, radiographic, and surgical evidence led to the diagnosis of OT. We collected demographic data and clinical characteristics of the patients, such as age, menstruation status, clinical symptoms, radiological investigations, laboratory examinations, operation findings, pathology, and outcome, by reviewing medical records. Pain intensity was measured on a scale of 0 (no pain) to 10 (extreme pain). We described scores of 1 to 3 as mild, 4 to 7 as moderate, and 8 to 10 as severe.

### 2.3. Outcome Measures

To identify the clinical risk factors of patients with ischemic ovary, we separated patients into two groups on the basis of the results of their operations: patients with ischemic OT and patients with non-ischemic OT. The primary outcome was the salpingo-oophorectomy rate. We constructed a clinical score on the basis of the logistic regression model’s final output.

### 2.4. Statistical Analysis

The results are presented as means, standard deviations, and the number of patients and rates. We compared the groups using univariate analyses. When appropriate, the chi-squared test and Fisher’s exact test were used for categorical variables, whereas the independent *t*-test was used for numerical variables. In addition, multivariate analyses were used to determine the independent predictors of ischemia OT. The statistical significance of numerical variables was determined using an ROC curve analysis. We used the Youden index to determine the appropriate cut-off point for ischemia OT diagnosis. The independent variable and all other essential variables in the univariate or ROC studies were incorporated as dependent variables in multivariate analyses utilizing logistic regression analysis. The final logistic regression model was used to create a clinical score, with one point assigned for the existence of each predictive predictor on the basis of the odds ratio. The risk factors’ sensitivity and specificity and the chances ratio of ischemia OT were investigated.

## 3. Results

During the study period, 125 patients were included. We excluded 17 from the study for prenatal ovarian torsion, leaving 118 patients with OT for analysis. Table 1 summarizes the demographic variables and characteristics associated with OT.

The participants’ mean age was 12.0 years (±4.3), and most were school-aged or adolescents (n = 112, 95%). Of the 118 enrolled, about two-thirds had experienced menarche. The median duration of the 118 patients was three days (IQR 1–5 days), ranging from 1 h to 10 days. The most common clinical symptoms were abdominal pain (96.7%) and vomiting (44.9%). About two-thirds of patients experienced moderate or severe pain on a visual pain scale. Before an operation, conservative treatment was suggested in 20 (16.9%) patients, and 16 (60%) of them had ischemic OT. At the time of operation, 78 (66.1%) of the patients had ischemic OT. Pelvic ultrasonography (US) with Doppler examination was the most commonly used imaging technique in 87 patients, and 24.1% showed diminished or absent flow on color Doppler imaging. Most (n = 85, 72%) patients arrived at the emergency room (ER), and 34.3% (n = 29) of patients were misdiagnosed. In total, 60 patients received US at the ER, and 17 (28.3%) had suspected ovarian torsion. Of the patients, 58 had computed tomography (CT) examinations at the emergency room (ER), and 25 (43.1%) had suspected ovarian torsion. The other patients who were shown to not have suggestive torsion by imaging were diagnosed with huge masses, tumors with hemorrhage, or, more rarely, malignancy on the US or CT reports. The mean interval between images and operation at ER was significantly longer in the imaging of patients without suspicion of torsion than in those with suspicion of torsion (31 h vs. 13.6 h, *p* = 0.022). Mean WBC was 11.2 ± 4.18 × 10^3^/mm^3^, and mean CRP was 33.8 ± 59.3 mg/L. Leukocytosis more significant than 10 × 10^3^/mm^3^ (52.6%, n = 70) and CRP more potent than 5 mg/L (44.5%, n = 40) were constant. The pelvic masses on preoperative ultrasound were 8.5 ± 4.0 cm, and 83.1% of them were statistically significant from 5 cm.

The comparison of clinical factors of OT in different outcomes is shown in Table 2. The univariate analyses showed that patients with ischemic OT were younger; had more frequent vomiting; and had higher WBC, CRP, and segments than patients with non-ischemic OT. In multivariate analyses, vomiting, elevated WBC, and elevated CRP were significant for ischemic OT.

Further ROC curve analysis found WBC ≧ 10 × 10^3^/mm^3^ or CRP ≧ 40 mg/L to be most notable for predicting ischemic OT. Following these findings, one point was assigned for the presence of vomiting and WBC ≧ 10 × 10^3^/mm^3^. Those with CRP ≧ 40 mg/L were assigned five points. The predictive power of each clinicopathological parameter is shown in Table 3.

A score of 1 or more represented a moderate risk for ischemic OT with a sensitivity of 92.7% and a specificity of 53.5%. Figure 1 shows the ROC curve for the clinical score in predicting the ischemic ovary torsion. The area under the curve was 0.85 (95% CI 0.76–0.94, *p* < 0.001).

## 4. Discussion

This study demonstrated several essential clinically relevant findings. First, ischemic OT was found to have a higher oophorectomy rate among pediatric OT patients. Second, the risk of ischemic ovary torsion increased when the OT patient experienced vomiting, leukocytosis, and elevated CRP. Additionally, we developed a predictive score for patients with ischemic OT.

Our study demonstrated that WBC and CRP count are essential factors in assessing the risk of ischemic adnexal torsion. These results are consistent with those of other studies and suggest that leukocytosis may indicate irreversible ischemic changes of a torted ovary [16] and increased CRP levels due to tissue necrosis and inflammation [17]. Compared with acute adnexal lesions where ovarian blood flow is not impaired, the tissue-degeneration process in OT can explain the higher CRP concentration [18]. In the present study, vomiting was significantly more frequent in patients with ischemic OT than in patients with non-ischemic OT. Vomiting may be triggered by the vagus nerve reflex secondary to pain [19]. This finding is in line with data from other studies that found that the presence of vomiting has a predictive value for the diagnosis of OT [18].

Some authors have speculated that, when the adnexal mass is 5 cm or larger, the risk of adnexal torsion increases [20]. Our study revealed that most pediatric children with ischemic OT did not have significantly larger masses than those with non-ischemic OT. Other laboratory examinations, including hemoglobin, were not inconstant and not significant between ischemic OT and non-ischemic OT patients. Previous papers revealed that patients’ age might be a crucial factor [21]. In our study population, patients with ischemic OT were potentially younger than patients with non-ischemic OT. In a retrospective study, premenarchal patients were compared with postmenarchal patients [21]. It was also more likely that a bluish-black ovary was found during surgery. Previous research has indicated that, in children under 15 years of age, normal ovaries have been demonstrated in 22 to 55% of patients with ovarian torsion [22,23].

In this study, one-third of cases had a preliminary missed diagnosis other than OT. These results match those observed in earlier studies finding that preoperative diagnosis for pediatric OT is difficult [24,25]. There are several possible explanations for this result. Pelvic US is the most commonly utilized imaging technique to evaluate patients in whom ovarian torsion is suspected [26]. In our study, only 28% were correctly diagnosed with torsion. Among those without Doppler examination, only 11% were accurately identified as torsion, while among those with Doppler examination, 81% were correctly diagnosed [27]. In our study, some patients had initially observed treatment for suspected spontaneous detorsion or mass resolution by the suggestion of pediatric or gynecologic surgeons, but 60% of them had ischemic OT at the time of operation. This suggestion was made according to earlier observations, which showed that 70% of adnexal masses in premenopausal patients would be resolved throughout several menstrual cycles [28], which may make the early diagnosis of ischemic OT in girls challenging.

The present study makes several noteworthy contributions to emergency care providers, pediatricians, pediatric surgeons, and gynecologic surgeons regarding the clinical characteristics and risk factors for ischemic OT in young patients. In addition, we developed the clinical score for ischemic OT. Our predictive OT score’s sensitivity was 92.7% with a score of ≧1, indicating that the OT score of this age group performs well.

We are aware of the study’s limitations, with retrospective single-center analyses being potentially biased as they does not include evaluations for other suspected diseases. However, up until now, there has been no prospective study predicting ischemia OT in the early stages. Furthermore, only surgically verified OT cases were included in this study. As a result, we cannot rule out the possibility that some individuals with spontaneous ovarian detorsion were not diagnosed with OT.

## 5. Conclusions

In summary, ovary torsion is a surgical emergency, and timely diagnosis is crucial. An exploratory laparoscopy should be performed as soon as possible in the presence of suspected OT with leukocytosis, vomiting, and CRP ≧ 40 mg/L. Further cross-national prospective studies are required for validation and for generalization to other ethnic backgrounds.

## Figures and Tables

**Figure 1 children-09-00206-f001:**
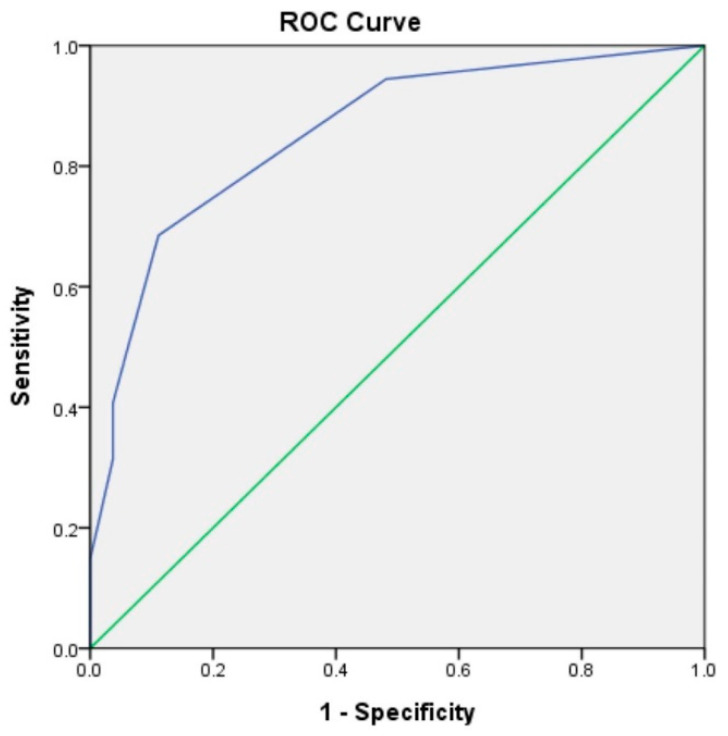
Receiver operating characteristic curve for clinical score in predicting the ischemic ovary torsion. The area under the curve was 0.85.

**Table 1 children-09-00206-t001:** Presenting signs and symptoms and descriptive statistics for patients with ovary torsion (N = 118).

	N = 118	
Age (years)	12 ± 4.3	
Menarcheal status		
Premenarchal	39	32.5%
Postmenarchal	81	67.5%
Mass size (cm)	8.7 ± 4.1	
≧4 cm	115	97.5%
≧5 cm	104	88.1%
Right site	71	60.2%
Abdominal pain	114	96.7%
Vomiting	53	44.9%
Fever	17	14.4%
Pain description, severity	114	
Mild (1–3)	50	43.8%
Moderate (4–7)	16	14.0%
Severe (>7)	48	42.1%
Duration of symptoms (D)	3.8 ± 5.8	
Ischemic ovary	78	66.1%

**Table 2 children-09-00206-t002:** Univariate and multivariate analyses between ischemic and non-ischemic ovary torsion.

	Univariate	Analysis		Multivariate Analysis	
	Ischemicn = 78	Non-Ischemic n = 40	*p*	adjusted OR	*p*
Age	12.2 ± 3.4	13.9 ± 3.1	0.009		
PreMC	39/78	9/40	0.081		
Duration	2.5 ± 1.7	2.4 ± 1.6	0.732		
Pain	76/78 (97.4%)	38/40 (95%)	0.603		
Moderate–severe Pain	44/78 (56.4%)	20/40 (50%)	0.508		
Fever	14/78 (17.9%)	3/40 (7.5%)	0.126		
Vomiting	43/78 (55.1%)	10/40 (25.0%)	0.002	4.025	1.027–15.772
Mass size	8.9 ± 4.2	8.3 ± 3.9	0.493		
HB, g/dL	12.5 ± 1.3	12.6 ± 1.5	0.605		
WBC, /mm^3^	12,383 ± 4146	8700 ± 2364	<0.001		
Leukocytosis	51/78 (65.4%)	11/40 (27.5%)	<0.001	1.000	1.000–1.001
Segment, %	77.1 ± 11.0	69.9 ± 11.7	0.002		
CRP, mg/L	50.4 ± 70.1	8.4 ± 16.4	<0.001	1.032	1.005–1.060
Hospital stay	4.4 ± 2.8	4.2 ± 3.0	0.710		
Salpingo-oophorectomy	68/78 (87.2%)	7/40 (17.5%)	<0.001		

**Table 3 children-09-00206-t003:** Sensitivity, specificity, NPV, and PVV of a predictive score for ischemic ovarian torsion.

	Sen	Spe	NPV	PPV
0	53.5	92.7	79.6	78.9
≧1	92.7	53.5	78.9	79.6
≧2	67.2	89.2	58.1	92.5
≧5	40.0	96.4	45.0	95.6
≧6	30.9	96.4	41.5	94.4
≧7	14.5	100	37.33	100

## Data Availability

All data are available from the corresponding author upon request.

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
