# Peer review of "Characteristics and Risk Factors for Ischemic Ovary Torsion in Children"

_children, 2022, doi:10.3390/children9020206_

Round 1
Reviewer 1 Report
Doppler examination was not sufficiently described and evaluated in the study.
There are some grammar errors on the file attached

Author Response
We would like to thank the reviewers for their extensive assessment of our manuscript, and for important and helpful comments and suggestions. We have taken all the remarks into account, in a manner that is described in detail below together with our answers to certain comments. We think that, following the reviewers’ suggestions, our manuscript has gained clarity and hope that the changes made will be considered satisfactory. The major changes are listed below:
Comment 1 Doppler examination was not sufficiently described and evaluated in the study.
Response: We thank the reviewer for this excellent suggestion. We have described the Doppler examination on the result section P3L113 as the following:
Pelvic ultrasonography (US) with Doppler examination was the most used imaging study in 87 patients, and 24.1% showed diminished or absent flow on color Doppler.
Comment 2 There are some grammar errors on the file attached
Response: We thank the reviewer for reminding us about grammar errors and had corrected them as the following:
“Institutional Review Board reviewed our hospital before the commencement of this study” corrected to P2L66 as the following:
“The Institutional Review Board of the Chang Gung Memorial Hospital approved the study.”
“This study demonstrated several essential clinically relevant finding findings.” Corrected to P5L145
“This study demonstrated several essential clinically relevant findings”
“First, ischemic OT is familiar with a higher oophorectomy rate among pediatric OT patients” Corrected to P5L146
“First, ischemic OT is familiar with a higher oophorectomy rate among pediatric OT patients.”
“However, up to now, there is no prospective study to early predict ischemic OT.”
Corrected to P6L190
“However, up to now, there is no prospective study to predict ischemia OT in the early stages.”

Reviewer 2 Report
My compliments to the authors for the extended retrospective analisys and the statistical interpretation of laboratory results.
My only concern is in results: n=29 patients were misdiagnosed. Sixty patients received ultrasonography (US), and 17 (28.3%) of them suspected ovarian torsion. Fifty-eight patients had computed tomography (CT) examinations, and 25(43.1%) of them suspected ovarian torsion.
29 patients misdiagnosed
17 patients suspected OT at US
25 patients suspected Ot at CT
a total of 71 patients.............and the other patients ????? Why an exploration was performed?????? the authors should report this topic.
An important bias of this study is that this is a retrospective study not including exploration performed for suspected other pathologies and as I sad before not reported.
Nevertheless the analysis is well performed and well described conclusions are correct according with the data analized
Author Response
We would like to thank the reviewers for their extensive assessment of our manuscript, and for important and helpful comments and suggestions. We have taken all the remarks into account, in a manner that is described in detail below together with our answers to certain comments. We think that, following the reviewers’ suggestions, our manuscript has gained clarity and hope that the changes made will be considered satisfactory. The major changes are listed below:
Comment 1 My compliments to the authors for the extended retrospective analisys and the statistical interpretation of laboratory results. My only concern is in results: n=29 patients were misdiagnosed. Sixty patients received ultrasonography (US), and 17 (28.3%) of them suspected ovarian torsion. Fifty-eight patients had computed tomography (CT) examinations, and 25(43.1%) of them suspected ovarian torsion. 29 patients misdiagnosed 17 patients suspected OT at US 25 patients suspected Ot at CT a total of 71 patients.............and the other patients ????? Why an exploration was performed?????? the authors should report this topic.
Response: We fully agree with the reviewer’s viewpoint and have reported on the result section on P3L118 as the following :
The other people who did not have suggestive torsion by image received exploration for huge mass, tumor with hemorrhage, or rarely malignancy on the US or CT reports. The mean interval between images and operation at ER was significantly longer in image non-suspected patients than in suspected patients (31 hr vs. 13.6 hr, P=0.022).
Comment 2 An important bias of this study is that this is a retrospective study not including exploration performed for suspected other pathologies and as I sad before not reported.
Response: We thank the reviewer for this excellent suggestion. We have added this on the discussion section P6L189 as the following:
We are aware of the study's limitations, including a retrospective single-center analysis with bias that does not include evaluation for other suspected diseases.
